# Engineering a single-chain immunoglobulin scaffold loaded with a latent-releasable cytotoxic pore-forming peptide

Izaskun Morillo [1], Joao Zulaica [1,2], Asier R. Caballero [1], Jaione Auzmendi-Iriarte [3], Eneko Largo [4], Beatriz Apellaniz [5,6], Arkaitz Carracedo [2,3,7,8,9], Marco Piva [3,7,8], José L. Nieva [1,2] ✉ & Edurne Rujas [1,6,8,10] ✉

Pore-forming peptides (PFPs) hold strong anti-tumor potential but require delivery systems to ensure stability and prevent off-target effects. In this study, we develop a strategy to transport a PFP in a biologically latent, precursor-like state. Specifically, based on the therapeutic antibody atezolizumab, which targets the tumor-associated programmed death-ligand 1, we engineered a single-chain IgG (scIgG) construct with flexible linkers embedding the melittin sequence. To further enable tumor-specific activation, melittin was flanked by regions cleavable by matriptase/ST14 (MT), a protease overexpressed in carcinomas. Initial constructs failed due to cytotoxicity during expression, prompting the redesign of melittin into a variant named Pmod2-2. This peptide retained potent pore-forming ability and was compatible with IgG fusion. The resulting scIgG-Pmod2-2 hybrid preserved Fab and Fc functionalities of the atezolizumab IgG, displayed favorable pharmacokinetics and released the active peptide in response to MT. These results highlight the potential of integrating cytolytic PFPs into antibody-based therapeutics.

Pore-forming peptides (PFPs)[1], *a.k.a.* anti-microbial peptides (AMPs)[2,3] or membrane-permeabilizing peptides (MPPs)[4], can be defined as ubiquitous small polypeptides (~10–40 residues) that interact with and permeabilize lipid bilayer membranes following different mechanisms[5]. Some PFPs show the capacity to exert potent in vitro and in vivo anti-tumor effects[4,6–8], and therefore constitute an emerging alternative to anti-cancer synthetic drugs[3,4,6,7]. Their cationic amphipathic structure enables them to target and damage the cell plasma membranes, implying that PFPs may display broad anti-cancer activity and a decreased probability of developing drug resistance. Moreover, the exerted membrane destabilization may facilitate uptake of cytotoxic drugs that enter cancer cells by passive diffusion[9].

However, to be clinically usable, PFPs need to combine potent and specific anti-cancer activity with a favorable pharmacokinetic profile[7]. Melittin, a naturally occurring peptide derived from bee venom, constitutes a PFP model lacking specificity for tumor cells[3,5,8,10–12]. Thus, despite its remarkable cytolytic efficacy and capacity for regulating the cell cycle, proliferation, angiogenesis, metastasis, and apoptosis of cancer cells[8,11], the clinical application of melittin is severely hindered owing to its indiscriminate toxicity and hemolytic activity. Development of melittin-based therapies to treat cancer, therefore, requires its target delivery to tumor sites[3]. In this regard, several approaches have been developed to attenuate the cytolytic-hemolytic activities of melittin and facilitate its safe delivery, including nanoparticle and hydrogel formulations, viral vectors, and as part of fusion proteins (reviewed in refs. [7,11,12]).

The amphipathicity of melittin drives its primary biological function, that is, its spontaneous water-membrane partitioning through hydrophobic and electrostatic interactions[4,8,10,13–15]. Thus, to increase the half-life of the

[1]Instituto Biofisika (UPV/EHU, CSIC), University of the Basque Country, Leioa, Spain. [2]Department of Biochemistry and Molecular Biology, University of the Basque Country (UPV/EHU), Bilbao, Spain. [3]CICbioGUNE, Derio, Spain. [4]Department of Immunology, Microbiology and Parasitology, Faculty of Medicine and Nursing, University of the Basque Country (UPV/EHU), Leioa, Spain. [5]Department of Physiology, Faculty of Pharmacy, University of the Basque Country (UPV/EHU), Vitoria-Gasteiz, Spain. [6]Bioaraba Health Research Institute, Microbiology, Infectious Disease, Antimicrobial Agents, and Gene Therapy, Vitoria-Gasteiz, Spain. [7]Traslational prostate cancer Research lab, CIC bioGUNE-Basurto, Biocruces Bizkaia Health Research Institute, Derio, Spain. [8]Ikerbasque, Basque Foundation for Science, Bilbao, Spain. [9]CIBERONC, Madrid, Spain. [10]Department of Pharmacy and Food Sciences, Faculty of Pharmacy, University of the Basque Country (UPV/EHU), Vitoria-Gasteiz, Spain. ✉e-mail: joseluis.nieva@ehu.eus; edurne.rujas@ehu.eus

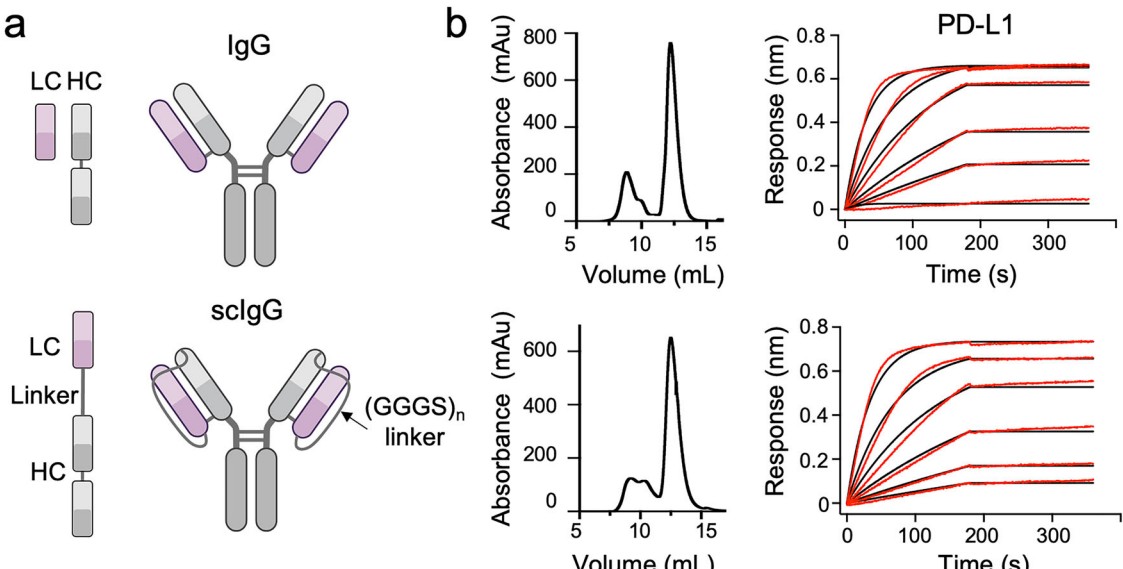

**Fig. 1 | Comparison between a single-chain IgG (scIgG) and a conventional IgG. a** Schematic representation illustrating the structural differences between a conventional IgG and a scIgG. In the scIgG design, the light chains (LC, depicted in pink) and heavy chains (HC, shown in gray) of the antibody are connected by flexible linkers, unlike the conventional IgG, where the LC and HC constitute separate chains. Created with BioRender.com. **b** Comparison of the SEC profiles obtained following affinity chromatography purification and binding kinetics to the PD-L1 receptor as measured by BLI of the conventional IgG and scIgG. A two-fold dilution series (25–0.8 nM) was used for the analysis. Red lines represent the experimental data, while black lines indicate the global fit from kinetic modeling.

peptide and to mitigate its hemolytic activity, some phospholipid-based delivery platforms transport melittin in a membrane-inserted state. These include liposomes[16], lipid-coated nanoparticles[17–20] and lipid nanodiscs[21–24]. However, while these approaches help stabilize melittin and reduce its toxicity, they do not ensure precise targeting to the site of action, potentially leading to off-target effects and reduced therapeutic efficacy.

To overcome these limitations and achieve controlled, specific PFP delivery, in this study, we leverage the single-chain IgG (scIgG) technology[25,26] to develop an innovative antibody-PFP hybrid. In this construct, both the N- and C-termini of the PFP are covalently fused to the antibody sequence, enabling the peptide to remain in a precursor-like, latent state during transport. We demonstrate that incorporating matriptase/ST14 (MT) cleavage sites flanking the PFP facilitates the controlled release of the active peptide, which can effectively disrupt lipid membranes. Initial attempts to incorporate the melittin sequence as the PFP in this format were unsuccessful, as they did not yield any viable product. Consequently, we designed a peptide variant, Pmod2-2, specifically optimized for compatibility with mammalian expression systems. Pmod2-2 was engineered to combine a potent cytolytic core with optimized protease cleavage sites, achieving a balance between portability and membrane-disrupting activity upon release.

For this design, we utilized the atezolizumab scaffold, an antibody targeting the tumor-associated programmed death-ligand 1 (PD-L1) approved for immune checkpoint blockade, to potentially enhance the PFP's anti-tumor activity. The inclusion of Pmod2-2 produced a hybrid molecule that retained binding capacity to its receptor, PD-L1, as well as to the neonatal Fc receptor (FcRn). As a result, the hybrid molecule exhibited an IgG-like half-life, lasting several days. This considerably extended the inherently short half-life of peptides, which are prone to rapid renal clearance due to their small size. The combination of prolonged circulation time and enhanced selectivity could enable sustained targeting of the PFP to the site of action, thereby broadening its therapeutic index.

## Results
### Design of an antibody scaffold for integrating cytotoxic sequences
Our main hypothesis establishes that an IgG scaffold can be engineered to incorporate a cytotoxic peptide in a latent form without compromising the

functional performance of its Fab and Fc domains and that, once delivered to the tumor via Fab recognition of a distinctive marker, this peptide could be selectively activated in the presence of specific proteases overexpressed in the tumor microenvironment. To test this hypothesis, we selected the atezolizumab IgG, an FDA-approved anti-PD-L1 antibody, used in the treatment of non-small-cell lung carcinoma, the most common type of lung cancer. We engineered a single-gene-encoded[25] atezolizumab by linking the carboxyl end of the light chain (LC) to the amino terminus of the heavy chain (HC) via a 70-aa flexible $(GGGS)_n$ linker (Fig. 1a). The resulting scIgG atezolizumab (scAtezo), expressed in HEK 293F cells, exhibited expression yields and PD-L1 binding profiles comparable to the conventional IgG (Fig. 1b).

Next, we further engineered the LC-HC linkers to incorporate the melittin sequence, flanked by the cleavage sites specific to matriptase/ST14 (MT)[27], a serine protease secreted in the tumor microenvironment of various cancers[28–30] (Fig. 2a). This modification was designed to enable enhanced control over peptide release. To identify potential challenges associated with its cytotoxicity, we also expressed a construct containing M70, a synthetically evolved derivative of the same size as melittin, but devoid of its cytotoxic activity[31]. The results, shown in Fig. 2b, confirmed that the cytotoxicity profiles of both sequences were replicated in relevant colorectal (COLO205), lung (H2221/H2228), and breast (MCF7) cancer cell lines, all of which have been previously demonstrated to be susceptible to the direct action of melittin[32–34].

Transfection of scIgG-peptide hybrid constructs into HEK 293 F cells revealed that scAtezo-melittin-transfected cells exhibited reduced viability compared to those transfected with scAtezo or scAtezo-M70, suggesting toxicity linked to the melittin sequence (Fig. 2c). Consistent with this, scAtezo-melittin cultures did not yield recoverable antibody (Fig. 2d). In contrast, the scAtezo-M70 hybrid was efficiently expressed and recovered in a functional form that bound PD-L1 (Fig. 2d, e).

In conclusion, these data demonstrate the feasibility of incorporating additional sequences into the scIgG scaffold via LC-HC linkers without compromising antibody binding. However, they highlight potential challenges in expressing cytotoxic sequences using this approach.

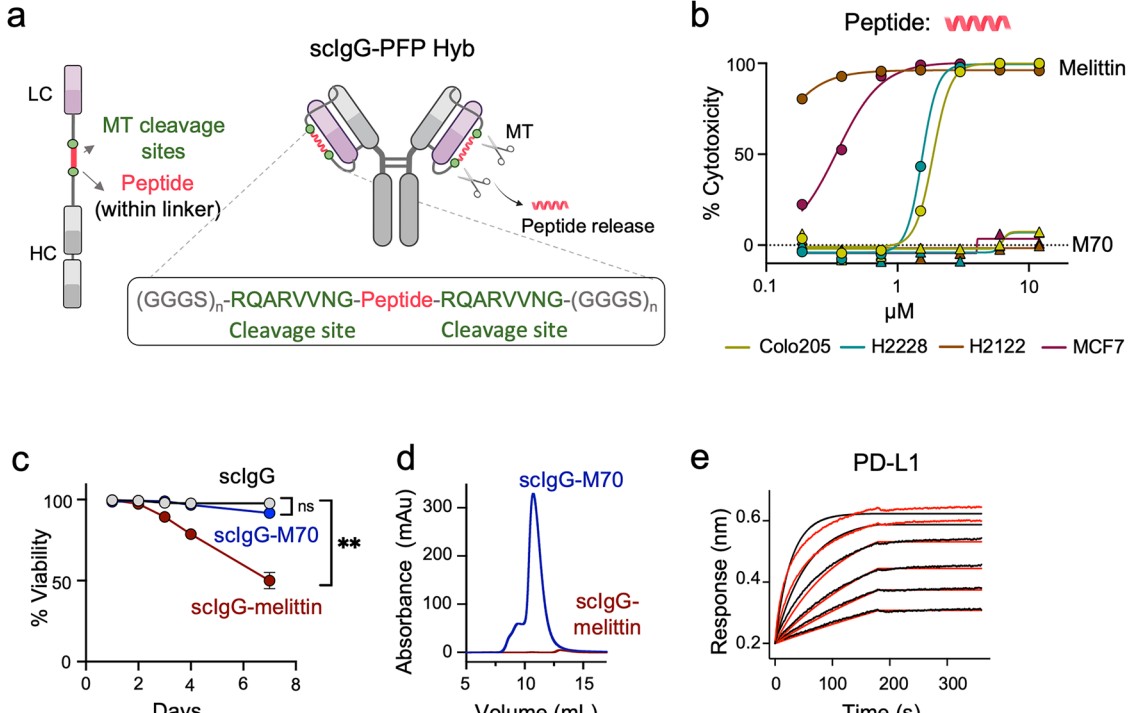

**Fig. 2 | The scIgG-PFP hybrid. a** Schematic representation of the integration of a peptide (melittin or M70) into the LC-HC linkers of scAtezo. The peptide (red) is flanked by cleavage sites for the MT ST14 (green), enabling its specific release under controlled conditions. Created with BioRender.com. **b** Dose-dependent cytotoxic activity of melittin (circle) against relevant cancer cell lines. M70 (triangle), a derivative of the same size but devoid of cytotoxic activity, was included as a negative control. One representative out of two independent replicates with similar results is shown. Mean values of two technical replicates are represented in the plot. **c** HEK 293 F cell viability assessed over the 7-day expression period, showing the impact of hybrid protein expression. Mean values ± SD of three independent replicates are represented in the plot. Statistical significance was determined using Welch's *t*-test. *p*-value = 0.0011 at day 7 (**p < 0.01). **d** SEC profiles of scAtezo-melittin (red, indicating non-expressed protein) and scAtezo-M70 (blue, showing proper folding). **e** Binding kinetics of scAtezo-M70 hybrid to PD-L1 measured using BLI. Red lines represent the experimental data, and black lines indicate the global fits.

## Redesign of peptides as cytolytic portable sequences

PFP sequences can be redesigned to optimize their functional activity[8,9,35]. Using M70 (producible/non-cytotoxic) and melittin (non-producible/cytotoxic) as reference sequences, we aimed to rationally design peptides with reduced cytotoxicity toward host cells, enabling their production as hybrid molecules while preserving their cytotoxic activity against tumor cells. To achieve this, we designed Pmod1 by introducing His residues, which are known to enhance PFP activity against cancer cells[9,35], at positions occupied by Glu and non-polar residues in M70 and melittin, respectively. Additionally, we increased the net hydrophobicity of the C-terminal helix by reintroducing the non-polar residues Ala (Pmod2) or Ala/Ile (Pmod3) from melittin, while preserving the amphipathicity of the N-terminal helix (Fig. 3a). The helical wheel representation and calculated values for mean hydrophobicity (H) and hydrophobic moment (μH) of these modified peptides[36] showed slightly higher values of these physicochemical parameters with respect to melittin, but overall similar profiles when compared to M70 (Fig. 3b, c).

COLO205 cell permeability assays, measured by flow cytometry (see Supplementary Fig. 1), demonstrated that the three peptides recovered the cell permeabilization activity of melittin. Among them, Pmod1 and Pmod2 exhibited comparable potency and were slightly more potent than Pmod3 (Fig. 3d). Based on the higher sequence similarity to our initial PFP target, melittin, we selected Pmod2 instead of Pmod1 for further study. First, we assessed Pmod2 potential for expression as a scIgG-peptide hybrid. As shown in Fig. 3e, the scAtezo-Pmod2 hybrid, incorporating the cytolytic peptide within the LC-HC linkers, was efficiently expressed in HEK 293 F cells. In contrast, constructs presenting Pmod2 at the termini of the IgG (either at the LC or HC C-terminus) could not be efficiently expressed. This likely reflects the increased propensity of the peptide to interact with

membranes when one of its ends is unconstrained, and/or difficulties in properly folding these hydrophobic sequences when added at those positions.

Despite the promising results for Pmod2, it is essential to address the impact of MT cleavage, as the released peptide will retain additional residues from the cleavage sites at both ends (Fig. 4a). To evaluate this, we synthesized Pmod2-1, a peptide mimicking the post-cleavage sequence, and observed that these extra residues reduced its cell-permeabilizing activity compared to Pmod2 (Fig. 4b). To mitigate this issue, we optimized the cleavage sites to produce a shorter peptide while maintaining its polarity and amphipathicity similar to those of melittin. The redesigned peptide, Pmod2-2, exhibited cell-permeabilizing potency comparable to Pmod2 (Fig. 4b). Circular dichroism (CD) spectra further confirmed that both peptides shared the potential to adopt α-helical structures in a low-polarity medium (Fig. 4c). Moreover, the physicochemical parameters of Pmod2-2, including charge, hydrophobicity (H), and hydrophobic moment (μH), closely aligned with those of Pmod2, confirming its structural and functional integrity (Fig. 4d). Additionally, transfection of recombinant scAtezo-Pmod2-2 into mammalian cells produced a properly folded and functional antibody (Fig. 4e), further validating the scIgG design as an effective scaffold for transporting PFP.

## Membranolytic activity of Pmod2-2

To assess the capacity of Pmod2-2 to directly permeabilize lipid membranes, we next evaluated its membranolytic activity in model vesicular systems. CD spectra in DPC micelles demonstrated that the peptide adopted an α-helical structure in this membrane-mimicking environment (Fig. 5a). Thus, we first performed solute leakage assays using vesicles made of POPC as a standard membrane model. The results revealed that Pmod2-2 effectively induced

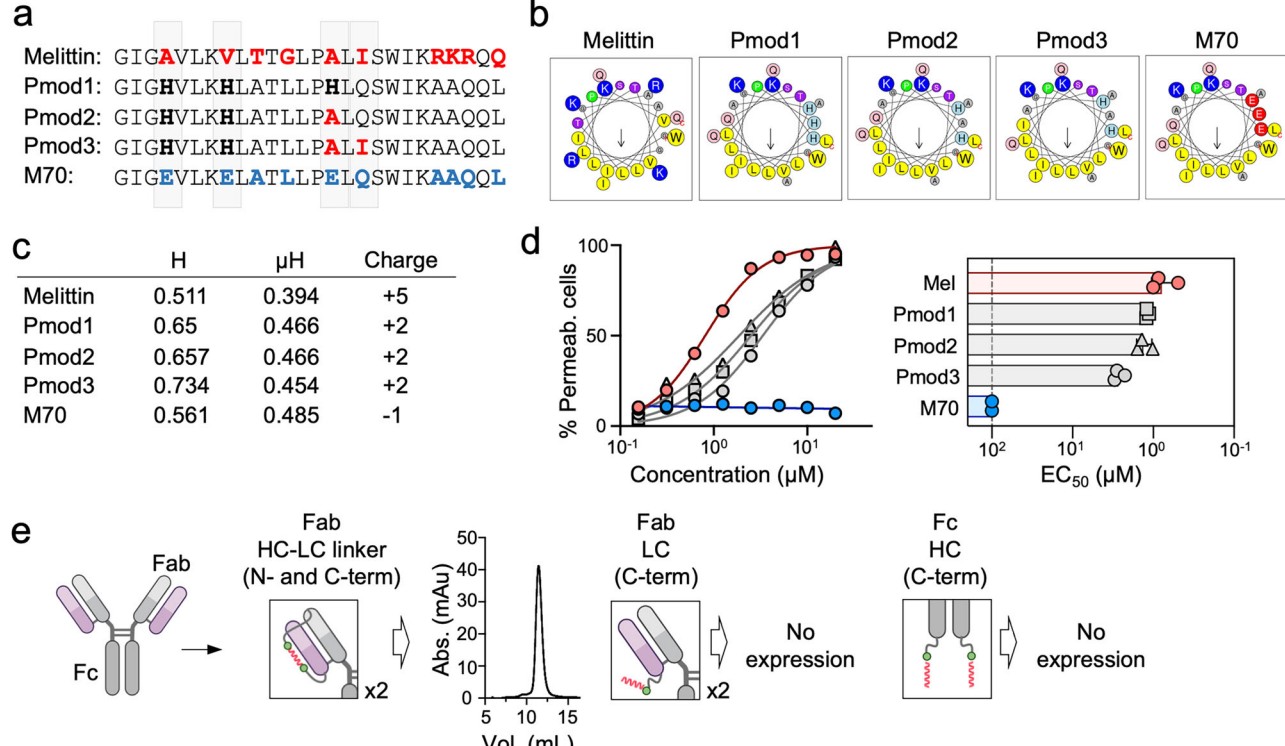

**Fig. 3 | Sequence modifications enabling cytolytic peptide transport within scAtezo. a** Sequences of modified peptides, using M70 as a reference for non-cytotoxic/producible peptides and melittin as a reference for cytotoxic/non-producible peptides. Highlighted in red are melittin residues, in blue M70 residues, and in bold black newly introduced residues. **b** Amphipathic properties of the peptides. Helical wheel representations rendered with the Heliquest server[36]. **c** Physicochemical parameters of the full sequences, including mean hydrophobicity (<H>); hydrophobic moment (<µH>) and charge, estimated using the same server[36].

**d** Representative titration experiment of COLO205 cell membrane permeabilization to 7-ADD after incubation with melittin, the modified peptides, or M70. One representative out of at least three independent replicates with similar results is shown. Mean values of two technical replicates are represented in the plot. Bar chart: $EC_{50}$ values determined from three independent experiments, with bars showing mean values. **e** Expression levels of the scAtezo-Pmod2 hybrid in HEK 293F cells with the peptide integrated at different positions: within the HC-LC linker, at the C-terminus of the LC, or at the C-terminus of the HC. Created with BioRender.com.

vesicle leakage, with the extent of the membrane permeability barrier disruption increasing with higher peptide concentrations (Fig. 5b). Additionally, Pmod2-2 permeabilized vesicles containing POPS and Chol. However, its activity was appreciably diminished in the presence of these lipids compared to POPC vesicles (Fig. 5c). This reduced activity likely arises from enhanced peptide-membrane electrostatic interactions with anionic POPS and from the more rigid nature of POPC:Chol membranes, both factors interfering with peptide translocation and hence imposing the requirement of higher peptide concentrations to achieve pore formation. These observations are consistent with previous studies on melittin, where negatively charged lipids or Chol were shown to inhibit pore assembly[15,37–39]. The same permeabilization pattern was observed for Pmod2 (Supplementary Fig. 2), suggesting that the extra residues added after proteolysis did not interfere substantially with the mechanism of pore formation.

Next, to investigate the pore size formed by Pmod2-2, we analyzed the permeabilization of single giant unilamellar vesicles (GUVs) using confocal microscopy (Fig. 5d and Supplementary Fig. 3). Permeabilization was assessed in the intact vesicles using fluorophores of varying molecular sizes, including Alexa 488 (Stokes radius ~0.5 nm) and FITC-labeled dextrans with molecular weights of 4 kDa (1.4 nm), 10 kDa (2.3 nm), 20 kDa (3.3 nm), and 40 kDa (4.5 nm). In POPC GUVs, Pmod2-2 formed pores with approximate radii between 1.4 and 2.3 nm, as judged from the efficient diffusion of the 4 kDa dextran, which was more restricted in the case of the 10 kDa dextran (Fig. 5e). Consistent with vesicle leakage assays (Fig. 5c), in POPC:Chol GUVs, higher peptide concentrations were required to observe vesicle permeabilization (250 nM compared to 100 nM in POPC). However, the resulting pores allowed substantial diffusion of 20 kDa dextran across POPC:Chol bilayers, while the 40 kDa dextran was mostly retained in the

lumen of the vesicles under the same conditions. This suggests formation of pores with radii between 3.3 and 4.5 nm, which are larger than those formed in POPC membranes (Fig. 5e).

The pore-forming activity of Pmod2-2 observed in vesicles was overall consistent with that described for melittin and some of its derivatives, requiring higher doses to permeabilize Chol-containing vesicles, but establishing therein pores permitting diffusion of larger molecules[31,39–41]. In conclusion, despite sequence differences, the scIgG-portable Pmod2-2 peptide exhibited a lytic activity profile in model membranes resembling that of melittin.

**Functional modulation of Pmod2-2 activity via the IgG scaffold**

Given the high sensitivity of Pmod2-2 to permeabilize POPC vesicles, we evaluated whether incorporation of the peptide within the linker, anchoring both ends, could suppress its activity and whether this activity could be restored upon MT incubation. Incubation of the purified scAtezo-peptide hybrids with MT resulted in specific cleavage, separating scAtezo into its HC and LC fragments, suggesting efficient peptide release from the IgG scaffold under controlled conditions. This proteolysis was also observed in the presence of PD-L1, indicating that Fab binding did not hinder access to the cleavage site (Fig. 6a and Supplementary Fig. 4a).

In the absence of MT, incubation of POPC LUVs encapsulating ANTS/DPX resulted in no detectable content release, suggesting that the lytic peptide remained constrained in its latent form. However, upon MT incubation, the released peptide led to vesicle permeabilization as evidenced by ANTS fluorescence emission (Fig. 6b). To determine the yield of peptide release by MT activity, equivalent amounts of Pmod2-2, either as a free synthetic peptide or integrated within the scAtezo-Pmod2-2 hybrid, were

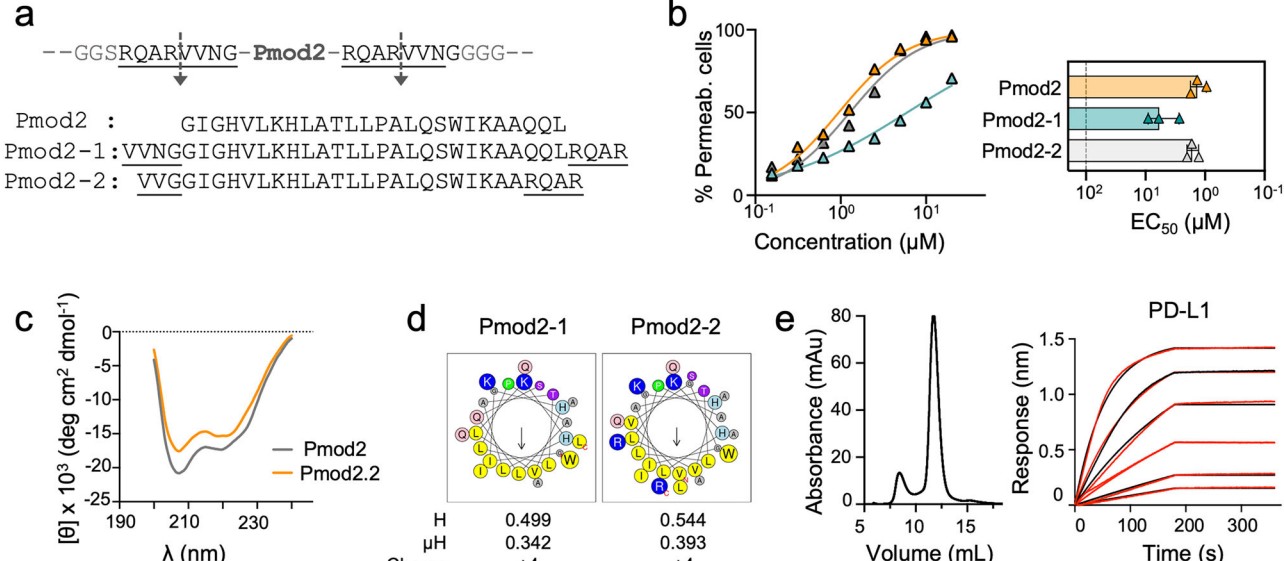

**Fig. 4 | Re-engineering of cleavage sites to preserve Pmod2 cytolytic activity.**
**a** Diagram showing residues added to Pmod2 following matriptase cleavage, along with sequences of Pmod2-based peptides. **b** COLO205 cell membrane permeabilization assays comparing the potency of Pmod2-1 (cyan) and Pmod2-2 (gray) with respect to Pmod2 (orange). One representative out of at least three independent replicates with similar results is shown. Mean values of two technical replicates are represented in the plot. Bar chart: $EC_{50}$ values ± SD determined from three independent experiments, with bars showing mean values. **c** CD spectra in HFIP and **d** helical wheel representations of Pmod2-1 and Pmod2-2. **e** SEC profile of purified scAtezo-Pmod2-2 and binding kinetics to PD-L1 analyzed by BLI. Red lines represent experimental data, and black lines show the global fit.

incubated in solution for 5 min, the latter in the presence of the enzyme. The lytic activity of the mixtures, subsequently assayed in LUVs, revealed that ca. 75% of Pmod2-2 was released from the scIgG-peptide hybrid through enzymatic cleavage (Fig. 6c). Importantly, vesicle permeabilization from the hybrid molecule was strictly dependent on the presence of MT, functional cleavage sites, and the specific peptide sequence. Control constructs lacking any of these critical components—such as scAtezo-Pmod2-2ΔMT (a scIgG-PFP hybrid without matriptase cleavage sites) and scAtezo (lacking the Pmod2-2 sequence)—failed to induce leakage (Fig. 6c). Furthermore, the scAtezo-Pmod2-2 remained unaltered upon incubation with cathepsin B, a protease upregulated in certain cancers that displays endopeptidase activity at neutral pH[42] (Supplementary Fig. 5). These findings underscore the precise and controlled activation of vesicle permeabilization mediated by MT activity.

**Functional properties conferred by the IgG Scaffold for Pmod2-2 delivery in vivo**

To assess the suitability of the antibody-peptide scaffold for systemic delivery of Pmod2-2 to tumor cells, we first compared the lytic activity of the free peptide and the scAtezo-Pmod2-2 hybrid over time in model membranes (Fig. 7a). The kinetic analysis of the scAtezo-Pmod2-2 hybrid revealed a baseline ANTS fluorescence signal during the 60-min incubation, further indicating that peptide release was strictly controlled and dependent on protease activity. Importantly, vesicle content release was nearly identical whether the hybrid was pre-incubated with LUVs for 1 h before MT addition or when MT and LUVs were added simultaneously, without prior incubation (established as 100% release) (Fig. 7a, upper panel). However, in the 1-h pre-incubation condition, a lag phase was observed, likely due to the time required for complete peptide digestion before full vesicle content release could occur (Supplementary Fig. 4b). In contrast, the free Pmod2-2 peptide exhibited a marked reduction in activity over time, with a 40% decrease after 5 min and an 80% decrease after 30 min incubation in buffer prior to addition of the vesicles, highlighting notable instability in solution (Fig. 7a, lower panel). These findings demonstrate that the antibody scaffold effectively protects and preserves peptide functionality, making it a reliable platform for targeted delivery. Thus, effective transport to the tumor via Fab

specificity would enable release of the cytotoxic peptide in close proximity to the target cell, facilitating its spontaneous insertion into the plasma membrane. Importantly, any premature release in circulation would be short-lived due to the peptide's low stability, thereby limiting off-target toxicity.

The ability of the scAtezo scaffold to extend the in vivo half-life of Pmod2-2 was evaluated in immunodeficient SCID mice. First, its interaction with FcRn in a pH-dependent manner was assessed using BLI. Notably, scAtezo-Pmod2-2 exhibited a higher binding affinity to FcRn than the standard scIgG ($K_D$ of 2.9 and 6.2 nM, respectively), which is likely attributable to interactions between the peptide and the receptor at acidic pH. Importantly, the pH-dependent binding profile, which is essential for FcRn-mediated antibody recycling, was maintained (Fig. 7b).

As expected, the molecule's ability to engage FcRn translated into a favorable pharmacokinetic profile. Incorporation of the peptide into the IgG scaffold enabled the hybrid molecule to exhibit IgG-like pharmacokinetics, remaining in circulation for over 10 days (Fig. 7c). Although peptide integration led to a slightly faster clearance, likely due to the introduction of hydrophobic residues into the IgG, this approach substantially extended the half-life of an otherwise rapidly eliminated peptide. Furthermore, no signs of toxicity or body weight loss were observed in the treated mice (Fig. 7d), further supporting the potential of this design to facilitate the systemic delivery of a latent peptide form while maintaining biocompatibility.

## Discussion

In this study, we successfully engineered a scIgG-based delivery system to integrate and transport PFPs, with a focus on the melittin-derived Pmod2-2 peptide. This design allows the peptide to remain in a biologically latent form during systemic circulation and to be specifically activated in the presence of MT, a serine protease overexpressed in carcinomas[28–30]. By combining precise targeting and conditional activation, this approach addresses two major limitations of PFP-based therapeutics: non-specific targeting, which can result in systemic toxicity, and poor pharmacokinetics, which reduce therapeutic efficacy[43,44].

One major challenge was the inherent toxicity of native melittin, which hindered its expression in mammalian cells when incorporated into the scIgG scaffold. This highlights a fundamental limitation in

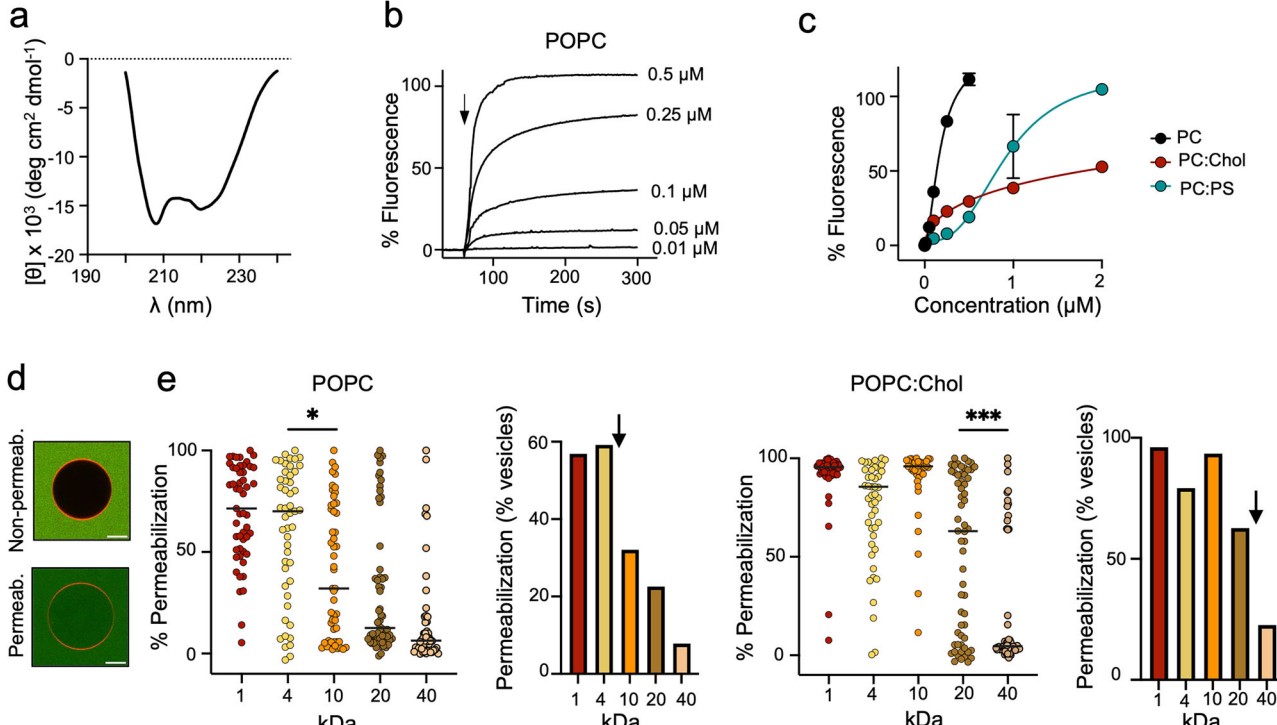

**Fig. 5 | Membrane-disruptive properties of Pmod2-2 across lipid compositions.** **a** CD spectra of Pmod2-2 in the membrane mimic DPC. **b** ANTS release kinetics following peptide addition (indicated by the arrow) at increasing concentrations to 100 μM POPC LUVs. **c** Permeabilization extents (% ANTS release after 5 min) measured in POPC (black), POPC:Chol (2:1, red), and POPC:POPS (2:1, cyan) LUVs as a function of the peptide concentration. Mean values ± SD of three independent replicates are shown. **d** Representative confocal micrographs of Rho-PE-labeled GUVs (orange) showing peptide-induced permeabilization (green). Scale bar 10 μm. **e** Solute-size-dependent permeabilization of POPC and POPC:Chol (2:1) GUVs. Dot plots show the distribution of solute entry (% permeabilization per vesicle) induced by Pmod2-2 (100 nM for POPC, 250 nM for POPC:Chol). Bars indicate median values of at least 50 GUVs. Bar charts show the percentage of vesicles exceeding 70% permeabilization, defining them as permeabilized. Black arrows indicate the estimated pore mean cut-size under each vesicle composition. Group differences were assessed using Welch's ANOVA followed by Games–Howell post-hoc tests. Significant pairwise differences are indicated in the figure. POPC 4 kDa vs 10 kDa $p$-value = 0.0353. POPC:Chol 20 kDa vs 40 kDa $p$-value = 0.0003. (*$p < 0.05$; ***$p < 0.001$).

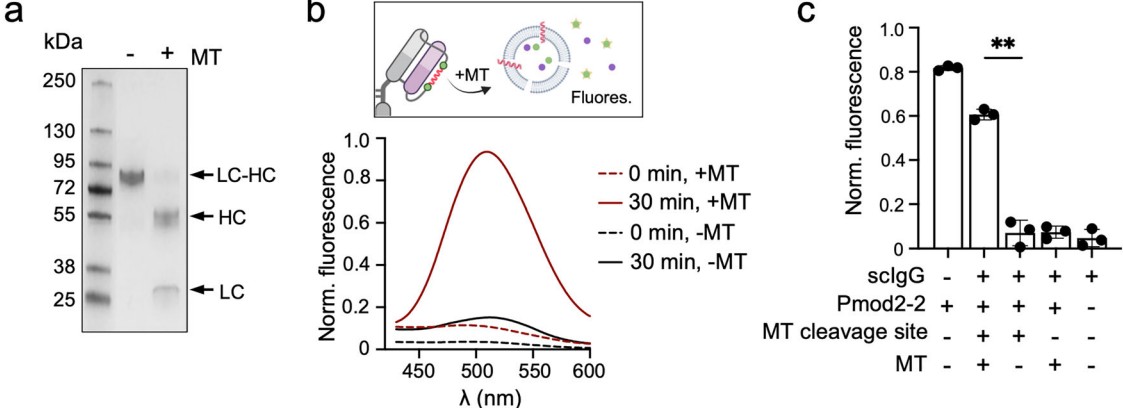

**Fig. 6 | Controlled proteolytic release of Pmod2-2 from scAtezo-Pmod2-2.** **a** SDS-PAGE analysis of the scIgG-Pmod2-2 hybrid incubated for 1 h with or without MT, demonstrating MT-mediated cleavage of the LC-HC fusion into distinct LC and HC fragments. **b** ANTS fluorescence spectra of 100 μM POPC LUVs loaded with ANTS/DPX, analyzed after direct addition of 0.15 mg/mL scIgG-Pmod2-2 and following 30 min of incubation in the presence or in the absence of MT. Leakage scheme created with BioRender.com. **c** Comparison of POPC LUV permeabilization following incubation with the Pmod2-2 peptide or the scIgG-Pmod2-2 hybrid, alongside the negative controls scAtezo (lacking Pmod2) and scAtezo-Pmod2-2ΔMT (lacking MT cleavage sites), in the presence or absence of metalloprotease (MT), as indicated in the panel. Mean values ± SD of $n = 3$ independent experiments. Differences were assessed using Welch's $t$-test ($p$-value = 0.0013, **$p < 0.01$).

developing antibody-based PFP delivery systems: the need for peptides that retain cytolytic activity while being compatible with protein expression platforms. To address this, we rationally designed the Pmod2-2 peptide, optimizing its sequence to balance cytolytic function and expression compatibility. Importantly, Pmod2-2 retained key functional attributes of melittin, such as its ability to form size-selective pores in lipid bilayers. Circular dichroism analysis and vesicle permeability assays further confirmed that Pmod2-2 maintained an α-helical structure in membrane-mimetic environments and exhibited a membranolytic profile comparable to that of melittin.

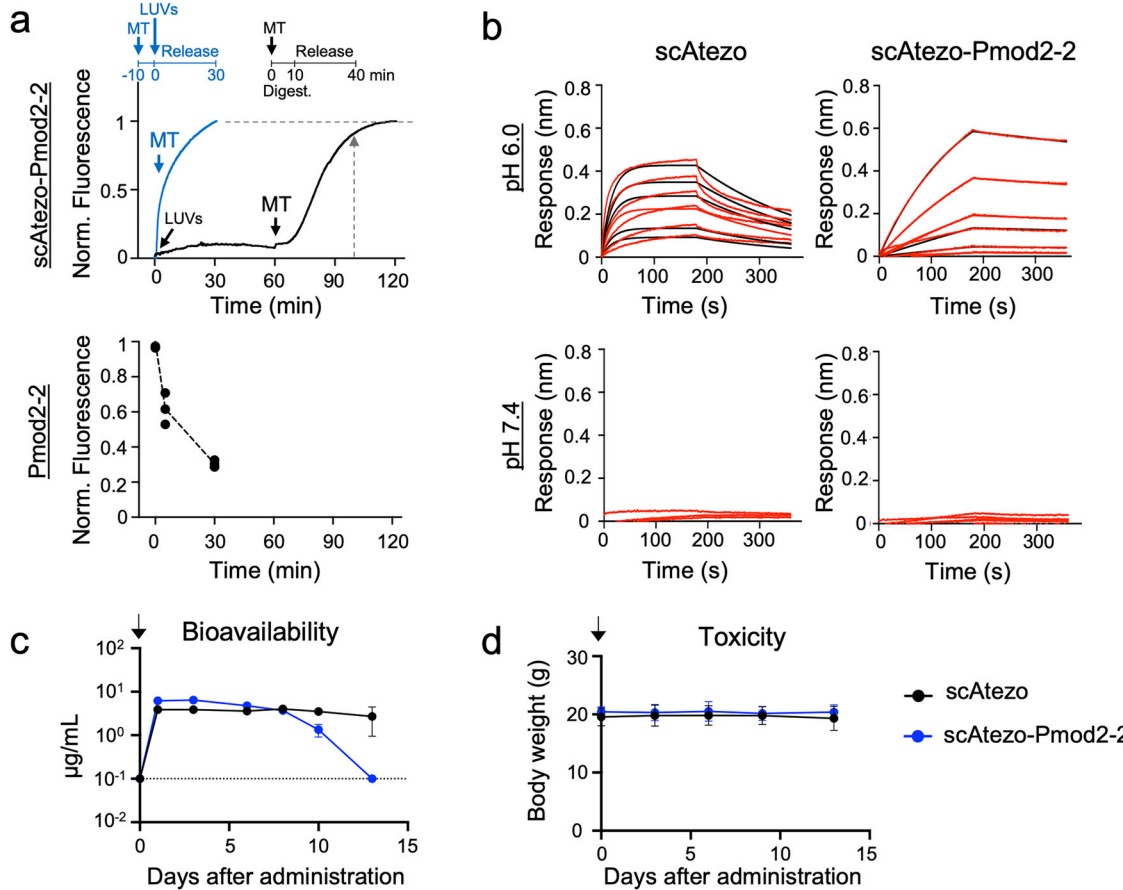

**Fig. 7 | Pmod2-2 stability and pharmacokinetic profile in the scIgG hybrid format. a** Comparison of the stability of Pmod2-2 as a free peptide or within the scIgG scaffold. Time trace of POPC LUVs permeabilization by Pmod2-2 released from the scIgG after MT addition (arrow) was assessed after 1 h of incubation with the vesicles (black curve) and upon simultaneous addition of the MT and the vesicles (blue curve), with a 10-min MT digestion allowed in both cases. ANTS release was normalized to the maximum release observed in the blue curve. ANTS release by Pmod2-2 in its free form was also measured. The ANTS release at *t* = 0 (no incubation) was set as the maximum activity, and the amount of ANTS released after 5

and 30 min of peptide incubation in solution, prior to vesicle addition, was measured. **b** Binding kinetics of scAtezo IgG and scAtezo-Pmod2-2 to FcRn at endosomal (pH 6) and physiological (pH 7.4) pH, using a two-fold dilution (25–0.8 nM). Red lines: raw data, black lines: global fits. **c** Serum levels of scAtezo-Pmod2 following intraperitoneal administration at a dose of 5 mg/kg (arrow) in SCID mice. scAtezo IgG was included as a reference. **d** Body weight changes in SCID mice after administration of 5 mg/kg (arrow) of each antibody. Data represent mean values ± SD for *n* = 3 mice in both (**c, d**).

Despite achieving the desired effect, the factors limiting the expression of cytotoxic peptides remain unclear. Melittin has a net charge of +5, concentrated at the C-terminus with a hydrophobic N-terminus, whereas Pmod2 has a lower, more evenly distributed charge (+2). While reducing the net positive charge through linker modifications might seem a plausible strategy, it is likely that overall hydrophobicity, net charge, and their distribution collectively influence expression feasibility. Further studies are therefore needed to explore how these properties balance membrane activity and expression compatibility. Such insights are critical for leveraging the modularity of this platform, which allows for the integration of alternative cytotoxic peptides, protease-cleavable linkers, and tumor-targeting antibodies, broadening its therapeutic potential for cancer treatment. Nevertheless, in the absence of detailed molecular knowledge relating sequence to portability, the adaptation of the scIgG platform to alternative cytotoxic cargos will require empirical validation.

A key feature of the scIgG-PFP design is the inclusion of MT-cleavable sites flanking the Pmod2-2 sequence within the scIgG linker, providing multifunctionality: targeted delivery, peptide inactivation, and controlled activation. In vitro vesicle assays confirmed the absence of non-specific permeabilization during the 1-h incubation of scAtezo-Pmod2-2 with LUVs, likely due to the constrained N- and C-termini of the peptide. This control over the peptide's inactive state appears to ensure safe systemic transport and minimize off-target toxicity, as indicated by the lack of body

weight loss in animals after administration of the hybrid. Furthermore, we have demonstrated that the inactive state can be reversed upon MT-mediated cleavage in vitro, where the addition of MT activates the peptide, enabling it to lyse lipid vesicles. This highlights the controlled mechanism of action of this therapeutic. However, to fully assess its potential, in vivo studies are needed to evaluate the system's ability to achieve localized activation at clinically relevant MT concentrations.

Due to its hydrophobic nature, Pmod2-2 tends to aggregate in solution. This was confirmed by the correlation between reduced activity and longer incubation times before vesicle addition, indicating that aggregation may limit its effectiveness. However, when incorporated into the scIgG scaffold, the peptide maintains its activity for longer periods of time, suggesting that the design improves peptide stability. A crucial aspect of our design is the demonstration that scAtezo-Pmod2-2 proteolysis proceeded efficiently in the presence of PD-L1, indicating that Fab binding to target antigen did not hinder MT access to the cleavage site (Supplementary Fig. 4a). Therefore, in the event of efficient transport and targeting by the engineered scIgG, peptide release will prospectively occur in close proximity to the target cell membrane, that is, after or during Fab engagement with the tumor marker. This proximity (in the order of several nm·s) will facilitate insertion of the cytolytic peptide into the membrane before the occurrence of sequence hydrophobicity-driven self-aggregation. Together, these features will ensure higher peptide concentrations at the tumor site, enhancing the action of the

antibody used for delivery, in this case, the immune checkpoint inhibitor atezolizumab. Nonetheless, comprehensive in vivo efficacy and safety studies in tumor-bearing models are essential to fully validate the system's potential and address any unforeseen toxicities.

Finally, the IgG scaffold provides key pharmacokinetic advantages. The scAtezo-Pmod2-2 hybrid binds to FcRn in a pH-dependent manner, facilitating recycling and extending the peptide's half-life, which, like other small peptides, would otherwise be prone to rapid renal clearance[43,44]. However, it is important to acknowledge that scAtezo-Pmod2-2 exhibits a faster decay rate compared to conventional IgG molecules, likely due to the inherent sequence properties of the incorporated peptide. Further optimization is required to improve this profile, potentially through masking strategies or sequence modifications that preserve activity while minimizing the introduction of hydrophobic or charged patches. Additionally, it will be crucial to assess and mitigate in vivo immunogenicity risks. De-immunization strategies should be explored to reduce the likelihood of anti-drug antibody formation, which could compromise the molecule's therapeutic potential.

In conclusion, this study presents a robust framework for antibody-based PFP delivery systems that combine targeted delivery with protease-mediated activation. By addressing peptide stability, lack of selectivity, and poor pharmacokinetics, our scIgG-Pmod2-2 platform offers a promising approach for precision cancer therapy. Future research on in vivo performance, protease specificity, and peptide optimization will unlock its full clinical potential.

## Methods
### Synthetic peptides, lipids, and fluorescent probes
The peptide sequences were produced as C-terminal carboxamides by ProteoGenix. Synthetic peptides were dissolved in dimethyl sulfoxide (DMSO; spectroscopy grade) and aliquots (typically 30 μL, 0.3 mM) were stored at −20 °C. 1-palmitoyl-2-oleoyl-glycero-3-phosphocholine (POPC), 1-palmitoyl-2-oleoyl-sn-glycero-3-phospho-L-serine (POPS), Cholesterol (Chol), and Rhodamine-1,2-dioleoyl-sn-glycero-3-phosphoethanolamine (Rho-PE) were purchased from Merck. Alexa Fluor 488 (thiol-reactive) was purchased from Thermo Fisher Scientific.

### Protein expression and purification
Genes encoding the atezolizumab IgG antibody, single-chain atezolizumab, and the single-chain atezolizumab fused to pore-forming peptides (scIgG-PFPs) were synthesized and cloned by GeneArt (Life Technologies) in the expression vector pcDNA3.4. 200 mL of HEK 293F cells (ATTC, USA) were seeded at a density of $0.8 \times 10^6$ cells/mL in Freestyle expression media and incubated with 135 rpm oscillation at 37 °C, 8% $CO_2$, and 70% humidity in a New Brunswick S41i incubator shaker (Eppendorf). Within 24 h of seeding, cells were transiently transfected using 50 μg of filtered DNA pre-incubated for 10 min at room temperature with the transfection reagent PEIPRO (Polyplus) at a 1:3 ratio. At day 7 post-transfection, cells were harvested by centrifugation at $6000 \times g$ and 4 °C for 10 min. The supernatant was filtered through a 0.22 μm Steritop filter (EMD Millipore) and purified using a Protein A affinity column (Cytiva). Fractions containing protein were pooled, concentrated, and loaded onto a Superdex 200 10/300 GL size-exclusion column (GE Healthcare) in 20 mM sodium phosphate buffer, pH 7.4, 150 mM NaCl (PBS). The FcRn and β2-microglobulin complex was produced by co-transfecting plasmids encoding each protein at a 1:1 ratio. For the human recombinant PD-L1 receptor, the expression vector pHLsec was used to produce a fusion construct with mVenus to enhance the solubility of the receptor. In both cases, transfection was performed as explained above for scIgG constructs. The resulting proteins were purified using a HisTrap Ni-NTA column (GE Healthcare) using a gradient elution of 500 mM Imidazole, 150 mM NaCl, 50 mM NaH2PO4, pH 7.4, followed by further refinement on a Superdex 200 10/300 GL size-exclusion column (GE Healthcare) equilibrated with PBS (pH 7.4).

### Binding assays
The binding affinities of scIgG-PFPs to PD-L1 were measured using bio-layer interferometry (BLI) on the Octet R8 system (Sartorius). Measurements were conducted in a kinetics buffer comprising PBS (pH 7.4) supplemented with 0.01% bovine serum albumin and 0.002% Tween, at 25 °C. Ni-NTA biosensors (Sartorius) were loaded with PD-L1 until a response signal of 0.8 nm was achieved. The biosensors were then transferred to wells containing a serial dilution of scIgG-PFPs, starting at 25 nM and decreasing to 0.8 nM, to measure the association phase. Dissociation rates were determined by dipping the biosensors into wells containing the kinetics buffer. Each step was conducted for 180 s. To assess the theoretical capacity of scIgG-PFPs for endosomal recycling, binding to the FcRn β2-microglobulin complex was measured at both physiological (pH 7.4) and endosomal (pH 6.0) conditions. Measurements were performed in PBS adjusted to the respective pH, supplemented with 0.01% bovine serum albumin and 0.002% Tween (pH 7.4) or 0.005% Tween (pH 6.0), at 25 °C. The FcRn β2-microglobulin complex was pre-immobilized onto Ni-NTA biosensors, and two-fold serial dilutions of scIgG-PFPs from 25 to 0.8 nM were used. Data analysis was performed with the Octet software (Sartorius) using a 1:1 binding model.

### Matriptase ST14 assays
Matriptase ST14 was purified from transformed *E. coli* BL21 cells (Thermo Fisher Scientific) grown in LB medium. Protein expression was induced with 0.5 mM IPTG at an $OD_{600}$ of 0.8–0.9, followed by a 4 h of incubation at 37 °C. The cells were harvested by centrifugation, and the pellet was resuspended in lysis buffer (50 mM Tris-HCl, pH 8.0, 500 mM KCl, 10% glycerol, 1 mM DTT). Cell lysis was performed using an Avestin Emulsiflex C5 homogenizer. After centrifugation at $10,000 \times g$ for 30 min, the pellet was solubilized in 50 mM Tris-HCl, pH 8.0, 500 mM KCl, 10% glycerol, 1 mM DTT, 6 M urea, and centrifuged again. Matriptase was purified using a HisTrap Ni-NTA column (GE Healthcare) in the presence of 6 M urea. The eluted fractions were dialyzed sequentially in two steps to gradually reduce urea concentration using buffers containing 50 mM Tris-HCl, pH 9.0, 10% glycerol, and 1 mM DTT. The sample was further purified by MonoQ ion-exchange chromatography, eluting with a gradient of 0–500 mM KCl in 50 mM Tris-HCl, pH 9.0, with 10% glycerol.

To assess cleavage of the peptide from the IgG scaffold, 2 μg of scIgG-PFP were incubated with 0.1 μg of Matriptase ST14 at 37 °C and 300 rpm for 1 h or 24 h. Following incubation, the reaction mixtures were analyzed by SDS-PAGE using a gel containing 12.5% acrylamide.

### Cell viability and permeability assays
The Colorectal cancer cell line COLO205 and the lung cancer cell lines NCI-H2122 and NCI-H2228 (all purchased from ATCC, USA) were cultured in RPMI 1640 medium (Merck) supplemented with 10% fetal bovine serum. The breast cancer cell line MCF7 (ATCC, USA) was cultured in DMEM medium (Merck) supplemented with 10% fetal bovine serum. For viability assays, 10,000 cells/well of each cell line were plated in 200 μL of medium and co-cultured with 2 μL of 2-fold serial dilutions of each peptide resuspended in DMSO. After 24 h of incubation at 37 °C, 5% $CO_2$, and 70% humidity, cell viability was assessed using the CellTiter-Glo 2.0 kit (Promega) according to the manufacturer's instructions. Luminescence, expressed as relative light units, was measured using 96-well white plates (Sigma-Aldrich) in a Synergy HT microplate reader (Biotek Instruments).

To measure the cell membrane permeability of COLO205 cells (ATCC, USA), 200 μL of cells were seeded at a density of 50,000 cells/well in 96-well plates. 2 μL of peptides ranging from 20 to 0 μM (2-fold dilutions) were added to the cells and incubated at 37 °C, 5% $CO_2$, and 70% humidity for 30 min. Then, plates were centrifuged at 1250 rpm for 5 min, the supernatant was discarded, and cells were resuspended in 200 μL of PBS containing 7-AAD (7-aminoactinomicina D, Thermo Fisher Scientific) according to the manufacturer's instructions. After 30-min incubation at 4 °C, percentages of cells that were permeable to 7-ADD were measured by flow cytometry in a CytoFlex Cytometer (Beckman Coulter).

## Circular dichroism

CD measurements were carried out on a thermally-controlled Jasco J-810 circular dichroism spectropolarimeter calibrated routinely with (1S)-$( + ) − 10$-camphorsulfonic acid, ammonium salt. Peptides were dissolved in an aqueous buffer (2 mM Hepes, pH 7.4) at 0.03 mM concentration with 25% hexafluoroisopropanol (HFIP) or 20 mM dodecylphosphocholine (DPC). Spectra were measured in a 1 mm path-length quartz cell equilibrated at 25 °C. Data were taken with a 1 nm bandwidth, 100 nm/min speed, and the results of 20 scans per sample were averaged.

## Vesicle preparation and membrane permeability assessment of large unilamellar vesicles

To prepare ANTS/DPX-encapsulating large unilamellar vesicles (LUVs) composed of POPC, POPC:POPS (1:1) or POPC:Chol (2:1), mixture lipids dissolved in chloroform were mixed and dried under vacuum to form a thin film, which was then resuspended in 12.5 mM ANTS, 45 mM DPX, 20 mM NaCl, and 5 mM Hepes. The vesicle suspension underwent 10 freeze-thaw cycles before extrusion through 0.1 μm Nucleopore polycarbonate filters to produce unilamellar vesicles of uniform size. Unencapsulated ANTS/DPX was removed by gel filtration chromatography. Permeabilization measurements were performed using an Aminco 8100 fluorimeter with excitation and emission wavelengths set to 355 nm and 520 nm, respectively. LUVs at a concentration of 100 μM were incubated with varying concentrations of peptides (0.01 μM to 5 μM) or 0.15 mg/mL scIgG-PFP, with or without MT ST14 treatment. To estimate maximal fluorophore leakage (100% release), 20 μL of Triton X-100 was added to the samples.

## Membrane permeability determined in single vesicles

Giant unilamellar vesicles (GUVs) composed of POPC or a POPC:Chol (2:1) mixture were prepared using the electroformation method[45]. Briefly, 2 mM total lipid, including 0.5% Rho-PE as a fluorescent membrane marker, was dissolved in 200 μL of chloroform. 2 μL of the solution was placed in the platinum wires of the chambers and connected to a function generator applying 10 Hz 2.5 V for 2 h 30 min and 2.5 Hz 2.5 V for 1 h for vesicle formation. The GUVs were transferred to a bovine serum albumin-blocked μ-Slide 8 Well Ibidi (Ibidi, Germany). Subsequently, 0.15 mM of Alexa Fluor 488 or different size FITC-Dextrans and 0.1 or 0.25 μM peptide were added to POPC GUVs or POPC:Chol GUVs, respectively, gently mixed, and incubated for 15 min before imaging. Confocal images were acquired using a Leica TCS SP5 microscope, and image analysis was performed with ImageJ software. Vesicle permeabilization percentage was calculated based on the intensity of Alexa Fluor 488 or FITC-Dextran inside the vesicles.

## Pharmacokinetics

In vivo pharmacokinetic studies were performed using 8-week-old C.B-17/IcrHan®Hsd-Prkdc$^{scid}$ female mice (3 mice/group, total of 6 mice) purchased from Innotiv (Envigo, license num.: TVWA/09/11452). This model was chosen because its immunodeficient background prevents the formation of anti-drug antibodies, allowing a more accurate pharmacokinetic assessment. Mice were housed in individually ventilated cages under a 12 h light/dark cycle at 21–23 °C and 40–55% humidity, with free access to food and water. Environmental enrichment (nesting material and shelters) was provided throughout the study. Animals were monitored daily for health status, body weight, and activity. Humane endpoints were defined as >20% body weight loss, persistent inactivity, or other signs of distress, in which case animals were euthanized by $CO_2$ inhalation followed by cervical dislocation. The procedures were minimally invasive, no analgesia was required, and no adverse events were observed during the study.

A single injection of 5 mg/kg of the antibodies was intraperitoneally injected, and blood samples were collected at multiple time points (days −1, 1, 3, 6, 8, 10, 13, 15) from the submandibular vein without anesthesia. Blood was maintained at room temperature for 30 min to allow coagulation, centrifuged at $1500 \times g$ for 10 min, and serum was collected and stored at −20 °C. At the experimental endpoint, mice were sacrificed by $CO_2$ inhalation followed by cervical dislocation.

Serum samples were assessed for levels of circulating antibodies by ELISA using standard curves for both scIgG and hybrid scIgG-PFP. Briefly, Nunc-Immuno™ MicroWell™ MaxiSorp 96-well plates (Thermo Fisher Scientific) were coated overnight with 2 μg/mL of human recombinant PD-L1. Then, a 1:100 serum dilution was incubated for 1 h at room temperature, and the antibody level was further detected using a goat anti-human-Fc secondary antibody bound to HRP (Thermo Fisher Scientific) at a 1:3000 dilution. The absorbance signal at 450 nm was quantified using a Synergy HT microplate reader (Biotek Instruments).

The experimental unit was a single mouse. The primary outcome measure was serum antibody concentration over time, which was used to determine antibody decay kinetics. The secondary outcome measure was body weight monitoring for toxicity. No inclusion or exclusion criteria were pre-specified. All animals enrolled in the study were included in the analyses, and no experimental units or data points were excluded. Sample size was not determined by formal power calculation; instead, group sizes were selected based on previous pharmacokinetic studies using similar mouse models and on ethical considerations to minimize animal use while ensuring robust detection of biologically relevant differences. Mice were randomly allocated to experimental groups according to body weight to ensure balanced distribution between groups. Outcome assessment and data analysis were not blinded due to the nature of the experiment.

## Statistics and reproducibility

Statistical analyses were performed using Prism version 8.0.1 (GraphPad Software Inc.). In GUV permeabilization data, group differences were assessed using Welch's ANOVA followed by Games–Howell post-hoc tests. Significant pairwise differences are indicated on the figure ($^*p < 0.05$; $^{***}p < 0.001$). Differences in LUV permeabilization by scAtezo-Pmod2-2, under both absence and presence of MT, were evaluated using Welch's $t$-test ($^{**}p < 0.01$). Welch's $t$-test was also applied to assess differences in cell viability following the expression of the hybrid constructs. Experiments were independently repeated at least three times unless otherwise indicated. For cell-based assays, n refers to the number of independent biological replicates (independent transfections/assays performed on different days). For biochemical and biophysical assays, n refers to independent protein preparations or vesicle batches. Technical replicates (e.g., duplicate or triplicate measurements within the same experiment) were averaged before calculating biological replicates. Data are presented as mean values ± SD, unless otherwise specified in the figure legends.

## Ethical approval statement

Animal protocols were approved by the Animal Research Ethics Board of CICbioGUNE (codes P-CBG-CBBA-0121 and P-CBG-CBBA-1321) and the Competent Authority (Diputación de Bizkaia) according to the guidelines of the European Union Council (Directive 2010/63/EU) and Spanish Government regulations (RD 53/2013). The Animal Facility at CICbioGUNE is accredited by AAALAC Intl., and the author has adhered to ARRIVE guidelines and complied with all relevant ethical regulations for animal use. A pre-registered protocol was not prepared for this study.

## Reporting summary

Further information on research design is available in the Nature Portfolio Reporting Summary linked to this article.

## Data availability

Data used to generate this manuscript are described in Supplementary Data 1. Reagents and additional information are available from the corresponding author upon reasonable request. Uncropped gels of Fig. 6a and Supplementary Fig. 4a are available in Supplementary Fig. 6 and Supplementary Fig. 7, respectively.

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

## Acknowledgements

The authors would also like to thank Dr. Ereño-Orbea and Dr. Jiménez-Barbero from CIC bioGUNE for providing access to the Octet R8 system and Dr. Julien from the Hospital for Sick Children for providing plasmids encoding receptors PD-L1, FcRn, and β2-microglobulin. This study was supported by grants PID2021-126014OB-I00, PID2021-122212OA-I00, PID2022-141200OA-I00, and PID2022-141553OB-I0 (FEDER/EU) funded by MCIN/AEI/10.13039/501100011033/FEDER, UE. Additional support was provided by the European Research Council (Consolidator Grant 819242). J.A. and M.P. acknowledge research contracts from the Ministry of Science and Innovation of Spain (JDC2023-052498-I funded by MICIU/AEI /10.13039/501100011033 and FSE+ and RYC2019-027726-I funded by MICIU/AEI /10.13039/501100011033 and El FSE invierte en tu futuro). I.M. and J.Z. acknowledge predoctoral contracts from the Basque Government, while E.R. received funding from the ERC (grant H2020-MSCA-COFUND-2020-101034228-WOLFRAM2).

## Author contributions

I.M., E.R., and J.L.N. conceived the study and designed the experimental approach. I.M. was responsible for the purification of antibodies and recombinant receptors and conducted all molecular biology procedures, confocal microscopy, fluorometric analyses, and cytotoxicity assays. A.R.C. provided assistance with the fluorometric experiments. J.A.I., M.P., and A.C. performed the in vivo mice work, including sample injections and blood collection. J.Z. performed the analysis of serum samples collected from the mice. B.A. and E.L. contributed to the circular dichroism and vesicle permeabilization assays. I.M., E.R., and J.L.N. carried out data analysis. E.R. and J.L.N. secured funding for the project. J.L.N. and E.R. drafted the manuscript with contributions from all authors.

## Competing interests

The authors declare no competing interests.
