## [Transparent Peer Review file · Communications Biology]

Engineering a single-chain immunoglobulin scaffold loaded with a latent-releasable cytotoxic pore-forming peptide

Corresponding Author: Dr Edurne Rujas

Version 0:

Reviewer comments:

Reviewer #1

(Remarks to the Author)

These authors have engineered an anti PDL 1 single chain antibody to contain protease cleavable linker sites to cytolytic pore forming peptides. Specifically, the authors engineered a flexible linker between the heavy chain and light chain with protease cleavable sites, so that pore forming peptides could be included in the single gene molecular design. They engineered a clever negative control with a melittin analog called M70 that is not cytotoxic. After finding that the original construct is too toxic to be produced in eukaryotic cells, the authors redesigned the pore forming peptide and the linkers iteratively. They ultimately found an altered melittin sequence that had the properties they had initially targeted.

This is an excellent paper! It has insightful data in areas ranging from Cancer cell biology to biophysics of peptides and membranes. In the end, the authors have been very clever and the results are a very encouraging step towards "smart" cancer therapy. This paper should be published, although perhaps the authors could address some minor comments below.

- 1) Do the authors have any information on nonspecific cleavage of their linker sequences by serum endo-proteases?
- 2) Can the authors discuss the serum stability of the released peptide. One would expect the peptide to be degraded, but perhaps not before the peptides act on local cells. It seems to me that this could be a huge benefit by reducing off target effects, if true.
- 3) The authors described the loss in activity in buffer for the peptide alone around line 328. This loss of activity is related to the solubility of the peptide. Could the authors discuss in more detail how peptide hydrophobicity would affect the peptide properties in vivo. Would increased hydrophobicity/decreased solubility cause the peptide to remain in the local tissue longer after release?

Reviewer #2

(Remarks to the Author)

This study presents a novel strategy for delivering pore-forming peptides (PFPs), using melittin as an example, in a biologically inactive form to reduce off-target toxicity. The authors engineered a single-chain IgG (scIgG) fusion protein embedding a modified melittin variant (Pmod2-2), flanked by protease-cleavable linkers responsive to matriptase, a tumor-associated enzyme. The redesigned peptide retained its cytolytic activity while being compatible with antibody fusion. The resulting construct remained stable, preserved target binding, and released the active peptide specifically in tumor environments.

This manuscript presents a well-conceived molecular design and a carefully executed experimental framework. The authors demonstrate a clear rationale behind the engineering of scAtezo-melittin, and the inclusion of a negative control (scAtezo-M70) is particularly commendable. This control strengthens the interpretation of the permeabilization activity and supports the specificity of the observed effects.

The study is innovative and addresses a relevant biological question with potential implications for therapeutic development. The experimental data are generally convincing, and the manuscript is well written. I believe the work will be of interest to the readership of Communications Biology.

However, I have a few minor comments that should be addressed to improve clarity and completeness:

i. Figure 2c:

Error bars are not shown in this figure, which limits the reader's ability to assess data variability and statistical significance. Please include error bars and clarify the statistical analysis performed.

ii. Figure 3d:

It is unclear why Pmod1 was not included or considered after this point. The rationale for selecting Pmod2 over Pmod1 should be clearly stated.

Reviewer #3

(Remarks to the Author)

The authors engineered a single-chain antibody (scAtezo) fused to a pore-forming peptide (Pmod2-2) that is selectively activated by tumor-overexpressed matriptase, thereby enabling safe delivery and controlled release of the peptide and overcoming the systemic toxicity, poor half-life, and off-target effects of conventional PFPs. While the work is innovative, I would like to discuss the following points with the authors:

- 1) How is off-target activation avoided in normal tissues expressing low levels of matriptase?
- 2) Is the scIgG platform restricted to pore-forming peptides, or can it be extended to other cytotoxic cargos?
- 3) The nucleotide sequences are not provided; could they be made available as supplementary data or in a public repository to facilitate validation and further engineering?

Version 1:

Reviewer comments:

Reviewer #1

(Remarks to the Author)

In this revision, the authors have expertly answered all of the critiques of the previous review, which were not significant criticisms, on any case. This revised version, in my opinion, should be published.

Reviewer #2

(Remarks to the Author)

I am satisfied with the authors' response to my comments and have no additional remarks.

Reviewer #3

(Remarks to the Author)

I have no further suggestions and recommend acceptance for publication.

Responses to Reviewers' comments:

Reviewer #1 (Remarks to the Author):

This is an excellent paper! It has insightful data in areas ranging from Cancer cell biology to biophysics of peptides and membranes.

We thank the reviewer for his/her encouraging comments.

Minor comments:

1) Do the authors have any information on nonspecific cleavage of their linker sequences by serum endo-proteases?

We designed the scAtezo hybrids carefully to avoid the presence of endoprotease recognition sites. Therefore, we expect that the expressed constructs are not substrates for serum endoproteases. Evidence supporting that peptide release only occurred upon specific cleavage with matriptase (MT), included the control experiments displayed in Figure 6c. In those experiments, we demonstrated that the scAtezo-Pmod2-2 construct devoid of protease cleavage sites remained unaltered upon incubation with the protease. To further support the requirement of specific MT activity, we include now an additional control experiment consisting in the incubation of the scAtezo-Pmod2-2 construct with cathepsin B, an endoprotease that is also upregulated in certain cancers (new supplementary Figure 5). Those results confirmed the absence of the hybrid scIgG cleavage upon incubation with cathepsin B, under conditions that otherwise led to efficient cleavage by MT. We have now added new text to discuss this finding in the revised version of the manuscript (lines 329-331).

2) Can the authors discuss the serum stability of the released peptide. One would expect the peptide to be degraded, but perhaps not before the peptides act on local cells. It seems to me that this could be a huge benefit by reducing off target effects, if true.

We agree with the reviewer that transporting the PFP in a latent/precursor-like state can prevent both its degradation in serum and off-target interactions. A crucial aspect of our strategy is that loading the PFP into scAtezo did not interfere with the specific binding to PD-L1 and FcRn by the functional scIgG moieties, Fab and Fc, respectively. Thus, in principle, peptide release would be confined to the local environment where MT activity is higher. Importantly, even if premature release occurred, the peptide's intrinsic instability in circulation would lead to rapid degradation, further minimizing the risk of systemic off-target toxicity. Following referee's recommendation, we have now discussed more extensively these aspects in the main text (see remodeled Abstract, and lines 349-352 in the revised manuscript).

3) The authors described the loss in activity in buffer for the peptide alone around line 328. This loss of activity is related to the solubility of the peptide. Could the authors discuss in more detail how peptide hydrophobicity would affect the peptides properties

in vivo. Would increase hydrophobicity/decreased solubility cause the peptide to remain in the local tissue longer after release?

We thank the reviewer for raising this issue, which was partially addressed in the response to the previous query. A crucial aspect of our design is the demonstration that proteolysis by MT proceeded efficiently in the presence of PD-L1, indicating that Fab binding to target antigen did not hinder access to the cleavage site (supplementary Fig. 4a). Therefore, the expectation is that peptide release will occur in close proximity to the target cell membrane, that is, after or during Fab engagement with the tumor marker. This proximity (in the order of several nm-s) will facilitate insertion of the cytolytic peptide into the membrane before occurrence of self-aggregation. Thus, in the event of efficient transport and targeting by the engineered scIgG, we do not expect a substantial formation of peptide aggregates in the local tissue. This is now included in the revised manuscript (Discussion section, lines 440-448).

Reviewer #2 (Remarks to the Author):

This manuscript presents a well-conceived molecular design and a carefully executed experimental framework... The study is innovative and addresses a relevant biological question with potential implications for therapeutic development.

We thank the reviewer for his/her positive assessment.

Minor comments:

i. Figure 2c: Error bars are not shown in this figure, which limits the reader's ability to assess data variability and statistical significance. Please include error bars and clarify the statistical analysis performed.

Following referee's indication, Figure 2c has been modified to represent the mean values \pm SD of 3 independent replicates.

ii. Figure 3d: It is unclear why Pmod1 was not included or considered after this point. The rationale for selecting Pmod2 over Pmod1 should be clearly stated.

We selected Pmod2 over Pmod1 for its higher resemblance to melittin sequence, the target PFP in our study. Following referee's instruction, we state now this in the text of the revised manuscript (lines 201-203).

Reviewer #3 (Remarks to the Author):

Points for discussion:

1) How is off-target activation avoided in normal tissues expressing low levels of matriptase?

As explained in the responses to the queries 2 and 3 of reviewer #1, our expectation is that Fab specificity would target the hybrid scIgG to the tumor tissue, in a way similar to clinically approved antibody-drug conjugates. We have modified the text to make this crucial aspect clearer in the revised manuscript (Abstract, and lines 349-352; 440-448 in the revised manuscript). We also note that by engineering the Fab component, the hybrid scIgG platform could in principle be adapted for the local delivery of PFPs into other kind of tumors (see lines 417-419 of the Discussion section).

2) Is the scIgG platform restricted to pore-forming peptides, or can it be extended to other cytotoxic cargos?

Similar to Fab specificity, the modularity of the scIgG design allows the inclusion of different cytotoxic cargos (lines 417-419 of the Discussion section). However, we caution that in the case of a different cargo, its sequence should be translated as a constituent section of the scIgG chain, and further allow the efficient expression of the complete construct in cells. In the absence of a detailed knowledge relating sequence and portability, this possibility should be empirically tested. We have added new text to the revised manuscript to discuss this issue (see Discussion section lines 420-422).

3) The nucleotide sequences are not provided; could they be made available as supplementary data or in a public repository to facilitate validation and further engineering?

Following reviewer's instructions, we have now included the nucleotide sequences of the constructs described in the paper in the supplementary data excel file.